# Multi-view Contrastive Learning for Entity Typing over Knowledge Graphs

**Zhiwei Hu**♣     **Víctor Gutiérrez-Basulto**◇     **Zhiliang Xiang**◇

**Ru Li**♣*     **Jeff Z. Pan**♠*

♣ School of Computer and Information Technology, Shanxi University, China
◇ School of Computer Science and Informatics, Cardiff University, UK
♠ ILCC, School of Informatics, University of Edinburgh, UK
♣ zhiweihu@whu.edu.cn,liru@sxu.edu.cn
◇{gutierrezbasultov,xiangz6}@cardiff.ac.uk
♠http://knowledge-representation.org/j.z.pan/

## Abstract

Knowledge graph entity typing (KGET) aims at inferring plausible types of entities in knowledge graphs. Existing approaches to KGET focus on how to better encode the knowledge provided by the neighbors and types of an entity into its representation. However, they ignore the semantic knowledge provided by the way in which types can be clustered together. In this paper, we propose a novel method called **M**ulti-view **C**ontrastive **L**earning for knowledge graph **E**ntity **T**yping (**MCLET**), which effectively encodes the coarse-grained knowledge provided by clusters into entity and type embeddings. MCLET is composed of three modules: i) *Multi-view Generation and Encoder* module, which encodes structured information from *entity-type*, *entity-cluster* and *cluster-type* views; ii) *Cross-view Contrastive Learning* module, which encourages different views to collaboratively improve view-specific representations of entities and types; iii) *Entity Typing Prediction* module, which integrates multi-head attention and a Mixture-of-Experts strategy to infer missing entity types. Extensive experiments show the strong performance of MCLET compared to the state-of-the-art.

## 1 Introduction

Knowledge graphs (KGs) (Pan et al., 2017a,b) store graph-like knowledge using triples of the form $(s, r, o)$, indicating that entities $s$ and $o$ are related to each other through a relation type $r$. KGs also contain entity type knowledge described as $(e, has\_type, t)$, denoting that entity $e$ has type $t$; e.g., we can express that *Joe Biden* has type *American_politician*, cf. Figure 1. Entity type knowledge plays a key role in various natural language processing related tasks, such as entity and relation linking (Gupta et al., 2017; Pan et al., 2019), knowledge graph completion (Peng et al., 2022; Niu et al., 2022; Wiharja et al., 2020), question

*Contact Authors

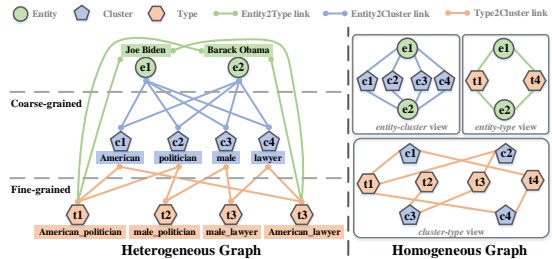

Figure 1: Heterogeneous graph with entities, coarse-grained clusters, and fine-grained types, covering three homogeneous views *entity-type*, *entity-cluster*, and *cluster-type*.

answering (Hu et al., 2022b; Chen et al., 2019; Hu et al., 2023), and relation extraction (Li et al., 2019). However, one cannot always have access to this kind of knowledge as KGs are inevitably incomplete (Zhu et al., 2015). For example, in Figure 1 the entity *Joe Biden* has types *American_politician* and *American_lawyer*, but it should also have types *male_politician* and *male_lawyer*. This phenomenon is common in real-world datasets, for instance, 10% of entities in the FB15k KG have the type */music/artist*, but are missing the type */people/person* (Moon et al., 2017). Motivated by this, we concentrate on the *Knowledge Graph Entity Typing* (KGET) task which aims at inferring missing entity types in a KG.

A wide variety of approaches to KGET have been already proposed, including embedding-(Moon et al., 2017; Zhao et al., 2020), transformer-(Wang et al., 2019; Hu et al., 2022a) and graph neural network (GNNs) based methods (Pan et al., 2021; Jin et al., 2022). Each of these approaches have their own disadvantages: embedding-based methods ignore the existing neighbor information, transformer-based are computationally costly due to the use of multiple transformers to encode different neighbors and GNNs-based ignore important higher-level semantic content beyond what is readily available in the graph-structure. Indeed,

in addition to relational and type neighbors, entity types can often be clustered together to provide coarse-grained information. Importantly, this type of coarse-grain information is available in many KGs, such as YAGO (Suchanek et al., 2007), which provides an alignment between types and Word-Net (Miller, 1995) concepts. The example in Figure 1 shows that between the entity *Joe Biden* and the type *American_politician*, there is a layer with cluster-level information *American* and *politician*. On the one hand, compared with the type content, the cluster information has a coarser granularity, which can roughly give the possible attributes of an entity and reduce the decision-making space in the entity type prediction task. On the other hand, the introduction of cluster information can enhance the semantic richness of the input KG. For example, from the type assertion (*Joe Biden*, *has_type*, *American_politician*) and the clusters *American* and *politician* corresponding to type *American_politician*, we can obtain new semantic connection edges between entities and clusters, *i.e.*, (*Joe Biden*, *has_cluster*, *American*) and (*Joe Biden*, *has_cluster*, *politician*). Note that as a type might belong to multiple clusters and a cluster may contain multiple types, a similar phenomenon occurs at the entity-type and entity-cluster levels, e.g, an entity might contain many clusters. Through the interconnection among entities, coarse-grained clusters and fine-grained types, a dense entity-cluster-type heterogeneous graph with multi-level semantic relations can be formed.

To effectively leverage the flow of knowledge between entities, clusters and types, we propose a novel method **MCLET**, a **M**ulti-view **C**ontrastive **L**earning model for knowledge graph **E**ntity **T**yping, including three modules: a *Multi-view Generation and Encoder* module, a *Cross-view Contrastive Learning* module and a *Entity Typing Prediction* module. The Multi-view Generation and Encoder module aims to convert a heterogeneous graph into three homogeneous graphs *entity-type*, *entity-cluster* and *cluster-type* to encode structured knowledge at different levels of granularity (cf. right side of Figure 1). To collaboratively supervise the three graph views, the *Cross-view Contrastive Learning* module captures the interaction by cross-view contrastive learning mechanism and mutually enhances the view-specific representations. After obtaining the embedding representations of entities and types, the *Entity Typing*

*Prediction* module makes full use of the relational and known type neighbor information for entity type prediction. We also introduce a multi-head attention with Mixture-of-Experts (MoE) mechanism to obtain the final prediction score. Our main contributions are the following:

- We propose MCLET, a method which effectively uses entity-type, entity-cluster and cluster-type structured information. We design a cross-view contrastive learning module to capture the interaction between different views.

- We devise a multi-head attention with Mixture-of-Experts mechanism to distinguish the contribution from different entity neighbors.

- We conduct empirical and ablation experiments on two widely used datasets, showing the superiority of MCLET over the existing state-of-art models.

Data, code, and an extended version with appendix are available at `https://github.com/zhiweihu1103/ET-MCLET`.

## 2 Related Work

**Embedding-based Methods.** These methods have been introduced based on the observation that the KGET task can be seen as a sub-task of the completion task (KGC). ETE (Moon et al., 2017) unifies the KGET and KGC tasks by treating the entity-type pair (*entity*, *type*) as a triple of the form (*entity*, *has_type*, *type*). ConnectE (Zhao et al., 2020) builds two distinct type inference mechanisms with local typing information and triple knowledge.

**Graph Neural Network-based Methods.** Given that GNNs inherently capture structural knowledge from graphs (e.g. the neighborhood of an entity), they have been previously used for the KGET task (Jin et al., 2019; Vashishth et al., 2020; Pan et al., 2021; Zhuo et al., 2022; Zhao et al., 2022; Zou et al., 2022a). For example, CET (Pan et al., 2021) introduces two mechanisms to fully utilize neighborhood information in an independent and aggregated manner. MiNer (Jin et al., 2022) proposes a neighborhood information aggregation module to aggregate both one-hop and multi-hop neighbors. However, these type of methods ignore other kind of semantic information, e.g., how types cluster together.

**Transformer-based Methods.** Many studies use transformer (Vaswani et al., 2017) for KGs related

tasks (Liu et al., 2022; Xie et al., 2022; Chen et al., 2022), including KGC. So, transformer-based methods for KGC, such as CoKE (Wang et al., 2019) and HittER (Chen et al., 2021) can be directly applied to the KGET task. TET (Hu et al., 2022a) presents a dedicated method for KGET. However, the introduction of multiple transformer structures brings a large computational overhead, which limits its application for large datasets.

## 3 Background

**Task Definition.** Let $\mathcal{E}$, $\mathcal{R}$ and $\mathcal{T}$ respectively be finite sets of *entities*, *relation types* and *entity types*. A *knowledge graph (KG)* $\mathcal{G}$ is the union of $\mathcal{G}_{triples}$ and $\mathcal{G}_{types}$, where $\mathcal{G}_{triples}$ denotes a set of triples of the form $(s, r, o)$, with $s, o \in \mathcal{E}$ and $r \in \mathcal{R}$, and $\mathcal{G}_{types}$ denotes a set of pairs of the form $(e, t)$, with $e \in \mathcal{E}$ and $t \in \mathcal{T}$. To work with a uniform representation, we convert the pair $(e, t)$ to the triple $(e, has\_type, t)$, where $has\_type$ is a special role type not occurring in $\mathcal{R}$. Key to our approach is the information provided by relational and type neighbors. For an entity $e$, its *relational neighbors* is the set $\mathcal{N}_r = \{(r, o) \mid (e, r, o) \in \mathcal{G}_{triples}\}$ and its *type neighbors* is the set $\mathcal{N}_t = \{(has\_type, t) \mid (e, has\_type, t) \in \mathcal{G}_{types}\}$. In this paper, we consider the *knowledge graph entity typing (KGET) task* which aims at inferring missing types from $\mathcal{T}$ in triples from $\mathcal{G}_{types}$.

**Type Knowledge Clustering.** Before we introduce our approach to KGET, we start by noting that it is challenging to infer types whose prediction requires integrating various pieces of information together. For example, to predict that the entity *Barack Obama* has type *20th-century_American_lawyer*, we need to know his birth year (*Barack Obama*, *was_born_in*, *1961*), place of birth (*Barack Obama*, *place_of_birth*, *Hawaii*), and occupation (*Barack Obama*, *occupation*, *lawyer*). Clearly, this problem is exacerbated by the fact that the KG itself is incomplete, which might more easily lead to prediction errors. However, in practice, type knowledge is often semantically clustered together, e.g., the types *male_lawyer*, *American_lawyer*, and *19th-century_lawyer* belong to the cluster *lawyer*. Naturally, this coarse-grained cluster information could help taming the decision-making process by paying more attention to types within a relevant cluster, without considering 'irrelevant' types from other clusters. With this in mind, we explore the intro-

duction of cluster information into the type prediction process. Therefore, a natural question is how to determine the clusters to which a type belongs to. In fact, the Freebase (Bollacker et al., 2008) and the YAGO (Suchanek et al., 2007) datasets themselves provide cluster information. For the Freebase dataset, the types are annotated in a hierarchical manner, so we can directly obtain cluster information using a rule-like approach based on their type annotations. For instance, the type */location/uk_overseas_territory* belongs to the cluster *location* and the type */education/educational_degree* belongs to the cluster *education*. The YAGO dataset provides an alignment between types and WordNet concepts[1]. So, we can directly obtain the words in WordNet (Miller, 1995) describing the cluster to which a type belongs to. For example, for the type *wikicategory_People_from_Dungannon*, its cluster is *wordnet_person_100007846*, and for the type *wikicategory_Male_actors_from_Arizona*, its cluster is *wordnet_actor_109765278*.

## 4 Method

In this section, we introduce our proposed method MCLET, which consists of three components: (1) *Multi-view Generation and Encoder* (§4.1); (2) *Cross-view Contrastive Learning* (§4.2); and (3) *Entity Typing Prediction* (§4.3).

### 4.1 Multi-view Generation and Encoder

For the KGET task, the two parts, $\mathcal{G}_{triples}$ and $\mathcal{G}_{types}$, of the input KG can be used for inference. The main question is how to make better use of the type graph $\mathcal{G}_{types}$, as this might affect the performance of the model to a large extent. So, the main motivation behind this component of MCLET is to effectively integrate the existing structured knowledge into the type graph. After introducing coarse-grained cluster information into the type graph, a three-level structure is generated: entity, coarse-grained cluster, and fine-grained type, such that the corresponding graph will have three types of edges: *entity-type*, *cluster-type*, and *entity-cluster*. Note that different subgraphs focus on different perspectives of knowledge. For example, the entity-cluster subgraph pays more attention to more abstract content than the entity-type subgraph. Therefore, to fully utilize the knowledge at each level, we convert the heterogeneous type graph into homogeneous graphs and construct an entity-type graph $\mathcal{G}_{e2t}$, a

---

[1]https://yago-knowledge.org/downloads/yago-3

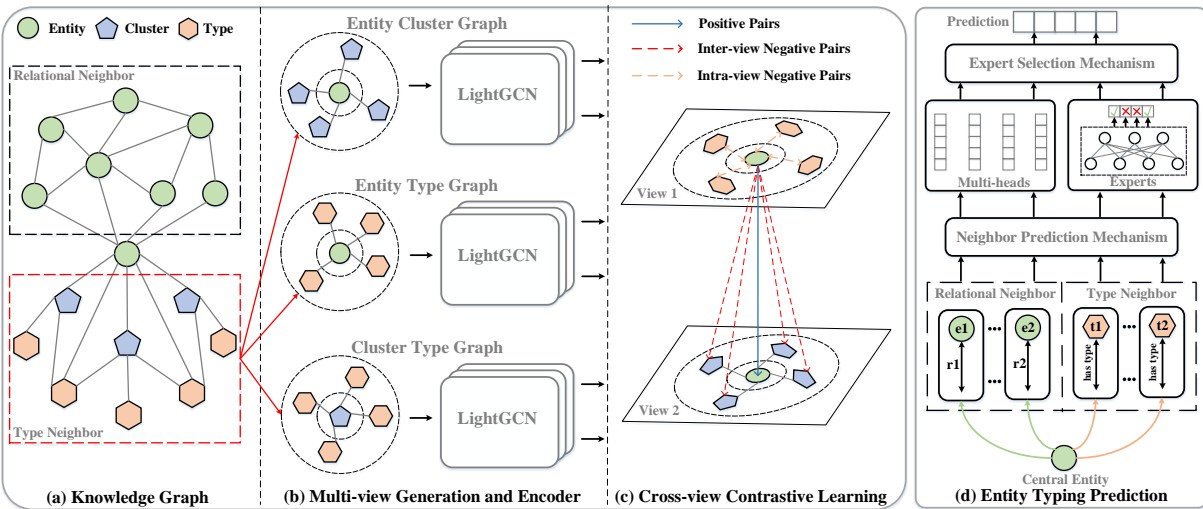

Figure 2: An overview of our MCLET model, containing three modules: Multi-view Generation and Encoder, Cross-view Contrastive Learning, and Entity Typing Prediction.

cluster-type graph $\mathcal{G}_{c2t}$, and a entity-cluster graph $\mathcal{G}_{e2c}$ separately.

**Entity Type Graph.** The entity type graph, denoted $\mathcal{G}_{e2t}$, is the original type graph $\mathcal{G}_{types}$ from the input KG. Recall that different types of an entity can describe knowledge from different perspectives, which might help inferring missing types. For example, given the type assertion *(Barack Obama, has_type, 20th-century_American_lawyer)*, we could deduce the missing type assertion *(Barack Obama, has_type, American_lawyer)*, since the type *20th-century_American_lawyer* entails *American_lawyer*.

**Cluster Type Graph.** The cluster type graph, denoted $\mathcal{G}_{c2t}$, is a newly generated graph based on how types are clustered. Type knowledge available in existing KGs inherently contains semantic information about clusters of types. For instance, the type */people/appointer* in FB15kET, clearly entails the cluster *people*. A similar phenomenon occurs in the YAGO43kET KG. Following this insight, for a type $t$ and its cluster $c$, we use a new relation type *is_cluster_of* to connect $t$ and $c$. For instance, from the type */people/appointer* and its cluster *people* we can obtain *(people, is_cluster_of, /people/appointer)*. Note that a type may belong to multiple clusters. For example, the type *American_lawyer*, belongs to the clusters *American* and *lawyer*.

**Entity Cluster Graph.** The entity cluster graph, denoted as $\mathcal{G}_{e2c}$, is generated based on $\mathcal{G}_{e2t}$ and $\mathcal{G}_{c2t}$. Unlike the entity type graph, the entity cluster graph captures knowledge at a higher level of abstraction. Therefore, its content has coarser granularity and wider coverage. So, given an entity $e$ and a type $t$, for a triple $(e, has\_type, t)$ from $\mathcal{G}_{e2t}$ and a triple $(c, is\_cluster\_of, t)$ from $\mathcal{G}_{c2t}$, we construct a triple $(e, has\_cluster, c)$, where *has_cluster* is a new relation type. Note that because a type may belong to multiple clusters, an entity with this type will also be closely related to multiple clusters. Consider for an example, the entity *Barack Obama* with type *American_lawyer*. Since *American_lawyer* belongs to the *American* and *lawyer* clusters, then there will be *(Barack Obama, has_cluster, American)* and *(Barack Obama, has_cluster, lawyer)* in $\mathcal{G}_{e2c}$.

**Multi-view Encoder.** We encode the different views provided by $\mathcal{G}_{e2t}$, $\mathcal{G}_{c2t}$, and $\mathcal{G}_{e2c}$ into the representations of entities, types and clusters using graph convolutional networks (GCN) (Kipf and Welling, 2017). More precisely, we adopt Light-GCN's (He et al., 2020) message propagation strategy to encode the information propagation from *entity-type*, *cluster-type*, and *entity-cluster* views. Our choice is supported by the following observation. The three graphs, $\mathcal{G}_{e2t}$, $\mathcal{G}_{c2t}$, and $\mathcal{G}_{e2c}$, are uni-relational, i.e., only one relational type is used, so there is no need a for a heavy multi-relational GCN model like RGCN (Schlichtkrull et al., 2018). Indeed, LightGCN is more efficient because it removes the self-connections from the graph and the nonlinear transformation from the information propagation function. To encode the three views, we use the same LightGCN structure, but no parameter sharing is performed between the correspond-

ing structures. Taking the encoding of $\mathcal{G}_{e2t}$ as an example, to learn the representations of entities and types, the $\ell$-th layer's information propagation is defined as:

$$
\begin{cases}
\mathbf{x}_{e\leftarrow e2t}^{(\ell)} = \displaystyle\sum_{t\in\mathcal{M}_e} \frac{1}{\sqrt{|\mathcal{M}_e||\mathcal{M}_t|}}\mathbf{x}_{t\leftarrow e2t}^{(\ell-1)} \\
\mathbf{x}_{t\leftarrow e2t}^{(\ell)} = \displaystyle\sum_{e\in\mathcal{M}_t} \frac{1}{\sqrt{|\mathcal{M}_t||\mathcal{M}_e|}}\mathbf{x}_{e\leftarrow e2t}^{(\ell-1)}
\end{cases} \quad (1)
$$

where $\{\mathbf{x}_{e\leftarrow e2t}^{(\ell)}, \mathbf{x}_{t\leftarrow e2t}^{(\ell)}\} \in \mathbb{R}^d$ represent the embeddings of entity $e$ and type $t$ in the graph $\mathcal{G}_{e2t}$, and $d$ is the dimension of embedding. $\{\mathbf{x}_{e\leftarrow e2t}^{(0)}, \mathbf{x}_{t\leftarrow e2t}^{(0)}\}$ are randomly initialized embeddings at the beginning of training. $\mathcal{M}_e$ and $\mathcal{M}_t$ respectively denote the set of all types connected with entity $e$ and the set of all entities connected with type $t$. By stacking multiple graph propagation layers, high-order signal content can be properly captured. We further sum up the embedding information of different layers to get the final entity and type representation, defined as:

$$
\mathbf{x}_{e\leftarrow e2t}^* = \sum_{i=0}^{L-1} \mathbf{x}_{e\leftarrow e2t}^{(i)}, \quad \mathbf{x}_{t\leftarrow e2t}^* = \sum_{i=0}^{L-1} \mathbf{x}_{t\leftarrow e2t}^{(i)} \quad (2)
$$

where $L$ indicates the number of layers of the Light-GCN. In the same way, we can get the type representation of cluster interaction $\mathbf{x}_{t\leftarrow c2t}^*$ and the cluster representation of type interaction $\mathbf{x}_{c\leftarrow c2t}^*$ from $\mathcal{G}_{c2t}$, and the entity representation of cluster interaction $\mathbf{x}_{e\leftarrow e2c}^*$ and the cluster representation of entity interaction $\mathbf{x}_{c\leftarrow e2c}^*$ from $\mathcal{G}_{e2c}$.

## 4.2 Cross-view Contrastive Learning

Different views can capture content at different levels of granularity. For example, the semantic content of $\mathbf{x}_{e\leftarrow e2c}^*$ in $\mathcal{G}_{e2c}$ is more coarse-grained than that of $\mathbf{x}_{e\leftarrow e2t}^*$ in $\mathcal{G}_{e2t}$. To capture multi-grained information, we use cross-view contrastive learning (Zhu et al., 2021; Zou et al., 2022b; Ma et al., 2022) to obtain better discriminative embedding representations. For instance, taking the entity embedding as an example, for the embeddings $\mathbf{x}_{e\leftarrow e2c}^*$ and $\mathbf{x}_{e\leftarrow e2t}^*$, our cross-view contrastive learning module goes through the following three steps:

**Step 1. Unified Representation.** We perform two layers of multilayer perceptron (MLP) operations to unify the dimension from different views as follows:

$$
\begin{aligned}
\mathbf{z}_{e\leftarrow e2t}^* &= \mathbf{W}_2(f(\mathbf{W}_1\mathbf{x}_{e\leftarrow e2t}^* + b_1)) + b_2 \\
\mathbf{z}_{e\leftarrow e2c}^* &= \mathbf{W}_2(f(\mathbf{W}_1\mathbf{x}_{e\leftarrow e2c}^* + b_1)) + b_2
\end{aligned} \quad (3)
$$

where $\{\mathbf{W}_1, \mathbf{W}_2\} \in \mathbb{R}^{d\times d}$ and $\{b_1, b_2\} \in \mathbb{R}^d$ are the learnable parameters, $f(\cdot)$ is the ELU nonlinear function. The embedding of the $i$-th entity in $\mathcal{G}_{e2t}$[2] can be expressed as $\mathbf{z}_{e_i\leftarrow e2t}^*$.

**Step 2. Positive and Negative Samples.** Let a node $u$ be an anchor, the embeddings of the corresponding node $u$ in two different views provide the positive samples, while the embeddings of other nodes in two different views are naturally regarded as negative samples. Negative samples come from two sources, intra-view nodes or inter-view nodes. Intra-view means that the negative samples are nodes different from $u$ in the same view where node $u$ is located, while inter-view means that the negative samples are nodes (different from $u$) in the other views where $u$ is not located.

**Step 3. Contrastive Learning Loss.** We adopt cosine similarity $\theta(\circ, \circ)$ to measure the distance between two embeddings (Zhu et al., 2021). For example, take the nodes $u_i$ and $v_j$ in different views, we define the contrastive learning loss of the positive pair of embeddings $(\boldsymbol{u}_i, \boldsymbol{v}_j)$ as follows:

$$
\begin{cases}
\mathcal{L}(\boldsymbol{u}_i, \boldsymbol{v}_j) = -\log \dfrac{e^{\theta(\boldsymbol{u}_i, \boldsymbol{v}_j)/\tau}}{\underbrace{e^{\theta(\boldsymbol{u}_i, \boldsymbol{v}_j)/\tau}}_{\text{positive pair}} + \underbrace{\mathcal{H}_{intra} + \mathcal{H}_{inter}}_{\text{negative pairs}}} \\
\mathcal{H}_{intra} = \displaystyle\sum_{k\in S_{intra}} e^{\theta(\boldsymbol{u}_i, \boldsymbol{u}_k)/\tau} \\
\mathcal{H}_{inter} = \displaystyle\sum_{k\in S_{inter}} e^{\theta(\boldsymbol{u}_i, \boldsymbol{v}_k)/\tau}
\end{cases}
$$
$$(4)$$

where $\tau$ is a temperature parameter, $\mathcal{H}_{intra}$ and $\mathcal{H}_{inter}$ correspond to the intra-view and inter-view negative objective function. If $u_i$ is in $\mathcal{G}_{e2t}$ and $v_j$ is in $\mathcal{G}_{e2c}$, then the positive pair embeddings $(\boldsymbol{u}_i, \boldsymbol{v}_j)$ represents $(\mathbf{z}_{e_i\leftarrow e2t}^*, \mathbf{z}_{e_j\leftarrow e2c}^*)$, i.e., the $i$-th node embedding in $\mathcal{G}_{e2t}$ and the $j$-th node embedding in $\mathcal{G}_{e2c}$ represent the same entity; after the contrastive learning operation, the corresponding node pair embedding becomes $(\mathbf{z}_{e_i\leftarrow e2t}^\diamond, \mathbf{z}_{e_j\leftarrow e2c}^\diamond)$. Considering that the two views are symmetrical, the loss of the other view can be defined as $\mathcal{L}(\boldsymbol{v}_j, \boldsymbol{u}_i)$. The final loss function to obtain the embeddings is the mean value of all positive pairs loss:

$$
\mathcal{L}_{\text{CL}} = \text{mean}(\sum_{i,j}[\mathcal{L}(\boldsymbol{u}_i, \boldsymbol{v}_j) + \mathcal{L}(\boldsymbol{v}_j, \boldsymbol{u}_i)]) \quad (5)
$$

## 4.3 Entity Typing Prediction

After performing the multi-view contrastive learning operation, we obtain two kinds of entity and

---

[2]We assume an arbitrary, but fixed order on nodes.

type representation. These two representations incorporate entities, coarse-grained clusters and fine-grained type knowledge at the same time. In this way, the cluster information is fully integrated into the representation of entities and types. For an entity $e$ and type $t$, we obtain their final representation by respectively concatenating $\mathbf{z}^{\diamond}_{e \leftarrow e2t}$ and $\mathbf{z}^{\diamond}_{e \leftarrow e2c}$, and $\mathbf{z}^{\diamond}_{t \leftarrow e2t}$ and $\mathbf{z}^{\diamond}_{t \leftarrow c2t}$:

$$\mathbf{z}_e = \mathbf{z}^{\diamond}_{e \leftarrow e2t} || \mathbf{z}^{\diamond}_{e \leftarrow e2c}, \ \ \mathbf{z}_t = \mathbf{z}^{\diamond}_{t \leftarrow e2t} || \mathbf{z}^{\diamond}_{t \leftarrow c2t} \quad (6)$$

**Neighbor Prediction Mechanism.** The entity and type embeddings are concatenated to obtain $\mathbf{z} = \mathbf{z}_e || \mathbf{z}_t$ as the embedding dictionary to be used for the entity type prediction task. We found out that there is a strong relationship between the neighbors of an entity and its types. For a unified representation, we collectively refer to the relational and type neighbors of an entity as neighbors. Therefore, our goal is to find a way to effectively use the neighbors of an entity to predict its types. Since different neighbors have different effects on an entity, we propose a neighbor prediction mechanism so that each neighbor can perform type prediction independently. For an entity, its $i$-th neighbor can be expressed as $(\mathbf{z}_i, \mathbf{r}_i)$, where $\mathbf{r}_i$ represents the relation embedding of the $i$-th relation. As previously observed (Pan et al., 2021), the embedding of a neighbor can be obtained using TransE (Bordes et al., 2013), we can then perform a nonlinear operation on it, and further send it to the linear layer to get its final embedding as follows: $\mathcal{N}_{(z_i, r_i)} = \mathbf{W}(\mathbf{z}_i - \mathbf{r}_i) + b$, where $\mathbf{W} \in \mathbb{R}^{N \times d}$, $b \in \mathbb{R}^N$ are the learning parameters, and $N$ represents the number of types. We define the embedding of all neighbors of entity $e$ as follows: $\mathcal{N}_e = [\mathcal{N}_{(z_1, r_1)}, \mathcal{N}_{(z_2, r_2)}, ..., \mathcal{N}_{(z_n, r_n)}]$, where $n$ denotes the number of neighbors of $e$.

**Expert Selection Mechanism.** Different neighbors of an entity contribute differently to the prediction of its types. Indeed, sometimes only few neighbors are helpful for the prediction. We introduce a **Multi-Head Attention** mechanism (Zhu and Wu, 2021; Jin et al., 2022) with a **Mixture-of-Experts** (**MHAM**) to distinguish the information of each head. We compute the final score as:

$$\begin{cases} \alpha_i = \phi(\mathbf{W}_2(\phi(\mathbf{W}_1 \mathcal{N}_e + b_1)) + b_2) \\ p = \sigma(\sum_{i=1}^{H} \phi(T_i \mathcal{N}_e \alpha_i) \mathcal{N}_e \alpha_i) \end{cases} \quad (7)$$

where $\mathbf{W}_1 \in \mathbb{R}^{M \times d}$, $\mathbf{W}_2 \in \mathbb{R}^{H \times M}$, $b_1 \in \mathbb{R}^M$ and $b_2 \in \mathbb{R}^H$ are the learnable parameters. $M$ and

$H$ respectively represent the number of experts in Mixture-of-Experts and the number of heads. $\phi$ and $\sigma$ represent the *softmax* and *sigmoid* activation functions respectively. $T_i > 0$ is the temperature controlling the sharpness of scores.

**Prediction and Optimization.** We jointly train the multi-view contrastive learning and entity type prediction tasks to obtain an end-to-end model. For entity type prediction, we adopt the false-negative aware (FNA) loss function (Pan et al., 2021; Jin et al., 2022), denoted $\mathcal{L}_{ET}$. We further combine the multi-view contrastive learning loss with the FNA loss, so we can obtain the joint loss function:

$$\begin{aligned} \mathcal{L}_{ET} &= -\sum_{(e_i, t_j) \notin \mathcal{G}_{e2t}} \beta(p_{i,j} - p_{i,j}^2) \log(1 - p_{i,j}) \\ &\quad - \sum_{(e_i, t_j) \in \mathcal{G}_{e2t}} \log p_{i,j} \\ \mathcal{L} &= \mathcal{L}_{ET} + \lambda \mathcal{L}_{CL} + \gamma ||\Theta||_2^2 \quad (8) \end{aligned}$$

where $\beta$ is a hyper-parameter used to control the overall weight of negative samples, $\lambda$ and $\gamma$ are hyper-parameters used to control the contrastive loss and $L_2$ regularization, and $\Theta$ is the model parameter set.

## 5 Experiments

| Datasets | FB15kET | YAGO43kET |
|---|---|---|
| # Entities | 14,951 | 42,335 |
| # Relations | 1,345 | 37 |
| # Types | 3,584 | 45,182 |
| # Clusters | 1,081 | 1,124 |
| # Train.triples | 483,142 | 331,686 |
| # Train.tuples | 136,618 | 375,853 |
| # Valid.tuples | 15,848 | 43,111 |
| # Test.tuples | 15,847 | 43,119 |

Table 1: Statistics of Datasets.

**Datasets.** We evaluate our MCLET model on two knowledge graphs, each composed of $\mathcal{G}_{triples}$ and $\mathcal{G}_{types}$. For $\mathcal{G}_{triples}$, we use the FB15k (Bordes et al., 2013) and YAGO43k (Moon et al., 2017). For $\mathcal{G}_{types}$, we use the FB15kET and YAGO43kET datasets introduced by (Pan et al., 2021), which map entities from FB15k and YAGO43k to corresponding entity types. The statistics of the corresponding datasets are shown in Table 1.

**Baselines.** We compare MCLET with three types of baselines. (*i*) embedding-based models: ETE (Moon et al., 2017), ConnectE (Zhao et al.,

| Datasets | FB15kET | | | | YAGO43kET | | | |
|---|---|---|---|---|---|---|---|---|
| Metrics | MRR | Hits@1 | Hits@3 | Hits@10 | MRR | Hits@1 | Hits@3 | Hits@10 |
| *Embedding-based methods* | | | | | | | | |
| ETE (Moon et al., 2017)◇ | 0.500 | 0.385 | 0.553 | 0.719 | 0.230 | 0.137 | 0.263 | 0.422 |
| ConnectE (Zhao et al., 2020)◇ | 0.590 | 0.496 | 0.643 | 0.799 | 0.280 | 0.160 | 0.309 | 0.479 |
| CORE (Ge et al., 2021)◇ | 0.600 | 0.489 | 0.663 | 0.816 | 0.350 | 0.242 | 0.392 | 0.550 |
| *GNN-based methods* | | | | | | | | |
| HMGCN (Jin et al., 2019)♦ | 0.510 | 0.390 | 0.548 | 0.724 | 0.250 | 0.142 | 0.273 | 0.437 |
| AttEt (Zhuo et al., 2022)◇ | 0.620 | 0.517 | 0.677 | 0.821 | 0.350 | 0.244 | 0.413 | 0.565 |
| ConnectE-MRGAT (Zhao et al., 2022)◇ | 0.630 | 0.562 | 0.662 | 0.804 | 0.320 | 0.243 | 0.343 | 0.482 |
| RACE2T (Zou et al., 2022a)◇ | 0.640 | 0.561 | 0.689 | 0.817 | 0.340 | 0.248 | 0.376 | 0.523 |
| CompGCN (Vashishth et al., 2020)♦ | 0.665 | 0.578 | 0.712 | 0.839 | 0.355 | 0.274 | 0.383 | 0.513 |
| RGCN (Pan et al., 2021)◇ | 0.679 | 0.597 | 0.722 | 0.843 | 0.372 | 0.281 | 0.409 | 0.549 |
| CET (Pan et al., 2021)◇ | 0.697 | 0.613 | 0.745 | 0.856 | 0.503 | 0.398 | 0.567 | 0.696 |
| MiNer (Jin et al., 2022)◇ | 0.728 | 0.654 | 0.768 | 0.875 | 0.521 | 0.412 | 0.589 | 0.714 |
| *Transformer-based methods* | | | | | | | | |
| CoKE (Wang et al., 2019)♦ | 0.465 | 0.379 | 0.510 | 0.624 | 0.344 | 0.244 | 0.387 | 0.542 |
| HittER (Chen et al., 2021)♦ | 0.422 | 0.333 | 0.466 | 0.588 | 0.240 | 0.163 | 0.259 | 0.390 |
| TET (Hu et al., 2022a)◇ | 0.717 | 0.638 | 0.762 | 0.872 | 0.510 | 0.408 | 0.571 | 0.695 |
| *Our methods* | | | | | | | | |
| MCLET-Pool | 0.726 | 0.644 | 0.773 | 0.881 | 0.524 | 0.418 | 0.589 | 0.715 |
| MCLET-MHA | 0.744 | 0.670 | 0.788 | 0.889 | 0.540 | 0.434 | 0.608 | 0.729 |
| MCLET-MHAM | **0.750** | **0.677** | **0.793** | **0.891** | **0.543** | **0.436** | **0.613** | **0.735** |

Table 2: Main evaluation results. ◇ results are from the original papers. ♦ results are from our implementation of the corresponding models. Best scores are highlighted in **bold**, the second best scores are underlined.

2020) and CORE (Ge et al., 2021); (*ii*) GNN-based models: HMGCN (Jin et al., 2019), AttEt (Zhuo et al., 2022), ConnectE-MRGAT (Zhao et al., 2022), RACE2T (Zou et al., 2022a), CompGCN (Vashishth et al., 2020) RGCN (Pan et al., 2021), CET (Pan et al., 2021), and MiNer (Jin et al., 2022); (*iii*) transformer-based models: CoKE (Wang et al., 2019), HittER (Chen et al., 2021) and TET (Hu et al., 2022a).

**Evaluation Protocol.** For every pair $(e, t)$ in the test set, we obtain a ranking list for the possible types $t$. We choose five automatic evaluation metrics: mean rank (MR), mean reciprocal rank (MRR), and Hits@$k$ ($k \in \{1, 3, 10\}$), MR measures the average positions of the first correct answer in a list of ranked results, MRR defines the inverse of the rank for the first correct answer, Hits@$k$ calculates the percentage of correct types ranked among the top-$k$, in addition to the MR metric, the larger the value, the better the effect. Following the evaluation protocol in most entity typing works (Pan et al., 2021; Hu et al., 2022a; Jin et al., 2022), all metrics are reported under the *filtered setting* (Bordes et al., 2013).

## 5.1 Main Results

The empirical results on entity type prediction are reported in Table 2. We can see that all MCLET

variants outperform existing SoTA baselines by a large margin across all metrics. In particular, compared to MiNer (the best performing baseline), our MCLET-MHAM respectively achieves 2.2% and 2.1% improvements on MRR in the FB15kET and YAGO43kET datasets. We can see a similar improvement e.g. on the Hits@1 metric, with an increase of 2.3% and 2.4% on FB15kET and YAGO43kET, respectively. Our ablation studies below show the contribution of MCLET's components on the obtained improvements.

We have evaluated three variants of MCLET to explore the effectiveness of the expert selection mechanism: MCLET-Pool, MCLET-MHA, and MCLET-MHAM. MCLET-Pool and MCLET-MHA respectively replace our expert selection mechanism with the pooling approach introduced in CET and the type probability prediction module introduced in MiNer. We observe that the MHAM variant achieves the best results. For instance, on FB15kET, MHAM improves 2.4% and 0.6% over the Pool and MHA variants on the MRR metric. This can be intuitively explained by the fact that the neighbors of an entity have different contributions to the prediction of its types. Indeed, by using the expert selection strategy, the information obtained by each head can be better distinguished. As a consequence, a more accurate final score can be

| Datasets | FB15kET | | | | | YAGO43kET | | | | |
|---|---|---|---|---|---|---|---|---|---|---|
| Setting | MRR | MR | Hits@1 | Hits@3 | Hits@10 | MRR | MR | Hits@1 | Hits@3 | Hits@10 |
| w/o e2t | 0.738 | 13 | 0.662 | 0.783 | 0.887 | 0.519 | 249 | 0.413 | 0.588 | 0.704 |
| w/o c2t | 0.745 | **12** | 0.670 | 0.788 | **0.891** | 0.540 | 170 | 0.433 | 0.609 | 0.731 |
| w/o e2c | 0.720 | 16 | 0.641 | 0.765 | 0.874 | 0.527 | 226 | 0.423 | 0.595 | 0.712 |
| w/o all | 0.683 | 20 | 0.601 | 0.726 | 0.843 | 0.470 | 498 | 0.365 | 0.535 | 0.658 |
| MCLET-MHAM | **0.750** | **12** | **0.677** | **0.793** | **0.891** | **0.543** | **167** | **0.436** | **0.613** | **0.735** |

Table 3: Evaluation of ablation experiments with different views on FB15kET and YAGO43kET. Best scores are highlighted in **bold**.

| Datasets | | FB15kET | | | | | YAGO43kET | | | | |
|---|---|---|---|---|---|---|---|---|---|---|---|
| Setting | | MRR | MR | Hits@1 | Hits@3 | Hits@10 | MRR | MR | Hits@1 | Hits@3 | Hits@10 |
| {all} | $l$=1 | 0.744 | **11** | 0.670 | 0.786 | 0.888 | **0.543** | **167** | **0.436** | **0.613** | **0.735** |
| | $l$=2 | 0.747 | 12 | 0.671 | 0.792 | **0.894** | 0.524 | 219 | 0.411 | 0.595 | 0.726 |
| | $l$=3 | 0.747 | 12 | 0.670 | **0.793** | **0.894** | 0.340 | 405 | 0.245 | 0.377 | 0.527 |
| | $l$=4 | **0.750** | 12 | **0.677** | **0.793** | 0.891 | 0.220 | 1368 | 0.160 | 0.237 | 0.330 |
| {1~4} | $l$=1 | **0.768** | 42 | **0.708** | **0.813** | **0.873** | **0.477** | **620** | **0.407** | **0.517** | **0.599** |
| | $l$=2 | 0.752 | 42 | 0.685 | 0.804 | 0.864 | 0.441 | 751 | 0.372 | 0.480 | 0.564 |
| | $l$=3 | 0.735 | **41** | 0.672 | 0.771 | 0.859 | 0.322 | 1125 | 0.267 | 0.350 | 0.419 |
| | $l$=4 | 0.674 | 48 | 0.603 | 0.712 | 0.809 | 0.297 | 1357 | 0.253 | 0.321 | 0.367 |

Table 4: Evaluation of ablation experiments with different LightGCN layers on FB15kET and YAGO43kET, where {all} indicates the complete dataset, and {1~4} indicates that the entities in the dataset only contain 1 to 4 type neighbors. Best scores are highlighted in **bold**.

obtained based on the prediction scores of each of the neighbors.

## 5.2 Ablation Studies

To understand the effect of each of MCLET's components on the performance, we carry out ablation experiments under various conditions. These include the following three aspects: a) the content of different views, see Table 3; b) different LightGCN layers, see Table 4; c) different dropping rates, see Table 5. Other ablation results and a complexity analysis can be found in Appendix B and C.

**Effect of Different Views.** We observe that removing any of the views most of the time will result in a decrease in performance, cf. Table 3. Further, if all three views are removed there will be a substantial performance drop in both datasets. For instance, the removal of all three views brings a decrease of 7.3% of the MRR metric on both datasets. This strongly indicates that the introduction of the three views is necessary. Intuitively, this is explained by the fact that each view focuses on a different level of granularity of information. Using the cross-view contrastive learning, we can then incorporate different levels of knowledge into entity and type embeddings. We can also observe that the performance loss caused by removing the cluster-type view is much lower than that caused by removing entity-type and entity-cluster views. This is

mainly because the cluster-type graph is smaller and denser, so the difference in the discriminative features of nodes is not significant.

**Effect of Different LightGCN Layers.** We have observed that the number of layers on the FB15kET and YAGO43kET datasets has a direct effect on the performance of LightGCN, cf. Table 4. For FB15kET, the impact of the number of GCN layers on the performance is relatively small. However, for YAGO43kET, the performance sharply declines as the number of layers increase. The main reason for this phenomenon is that in the YAGO43kET dataset most entities have a relatively small number of types. As a consequence, the entity-type and entity-cluster views are sparse. So, when deeper graph convolution operations are applied, multi-hop information is integrated into the embeddings through sparse connections. As a consequence, noise is also introduced, which has a negative impact on the final results. We further constructed a dataset where each entity contains between 1 to 4 type neighbors and performed experiments with LightGCN with layer numbers ranging from 1 to 4. In this case, we can observe that as the number of GCN layers increase, there is a significant decline in the performance on FB15kET as well. This is in line with the finding that the performance decreases in YAGO43kET as the number of layers increases.

**Effect of Dropping Rates of Relation Neighbors.**

| Dropping Rates | 25% | | | 50% | | | 75% | | | 90% | | |
|---|---|---|---|---|---|---|---|---|---|---|---|---|
| Models | MRR | H@1 | H@3 | MRR | H@1 | H@3 | MRR | H@1 | H@3 | MRR | H@1 | H@3 |
| CompGCN (Vashishth et al., 2020) | 0.661 | 0.573 | 0.705 | 0.655 | 0.565 | 0.702 | 0.648 | 0.559 | 0.697 | 0.633 | 0.544 | 0.679 |
| RGCN (Pan et al., 2021) | 0.673 | 0.590 | 0.716 | 0.667 | 0.584 | 0.708 | 0.648 | 0.560 | 0.694 | 0.626 | 0.534 | 0.673 |
| CET (Pan et al., 2021) | 0.697 | 0.613 | 0.744 | 0.687 | 0.601 | 0.733 | 0.670 | 0.580 | 0.721 | 0.646 | 0.553 | 0.698 |
| TET (Hu et al., 2022a) | 0.712 | 0.631 | 0.758 | 0.705 | 0.624 | 0.753 | 0.689 | 0.606 | 0.733 | 0.658 | 0.574 | 0.701 |
| MiNer (Jin et al., 2022) | 0.714 | 0.634 | 0.760 | 0.703 | 0.620 | 0.749 | 0.683 | 0.596 | 0.731 | 0.652 | 0.556 | 0.706 |
| MCLET-MHAM | **0.742** | **0.665** | **0.788** | **0.732** | **0.653** | **0.777** | **0.718** | **0.636** | **0.765** | **0.700** | **0.614** | **0.751** |

Table 5: Evaluation with different relational neighbors dropping rates on FB15kET. H@N is an abbreviation for Hits@N, $N \in \{1, 3\}$. Best scores are highlighted in **bold**.

| Dropping Rates | 25% | | | 50% | | | 75% | | | 90% | | |
|---|---|---|---|---|---|---|---|---|---|---|---|---|
| Models | MRR | H@1 | H@3 | MRR | H@1 | H@3 | MRR | H@1 | H@3 | MRR | H@1 | H@3 |
| CompGCN (Vashishth et al., 2020) | 0.664 | 0.578 | 0.708 | 0.662 | 0.574 | 0.708 | 0.653 | 0.565 | 0.699 | 0.637 | 0.546 | 0.683 |
| RGCN (Pan et al., 2021) | 0.676 | 0.593 | 0.719 | 0.673 | 0.590 | 0.719 | 0.658 | 0.573 | 0.702 | 0.636 | 0.548 | 0.681 |
| CET (Pan et al., 2021) | 0.699 | 0.617 | 0.743 | 0.694 | 0.610 | 0.742 | 0.675 | 0.588 | 0.721 | 0.653 | 0.564 | 0.700 |
| TET (Hu et al., 2022a) | 0.711 | 0.631 | 0.756 | 0.710 | 0.630 | 0.757 | 0.690 | 0.608 | 0.734 | 0.677 | 0.591 | 0.722 |
| MiNer (Jin et al., 2022) | 0.716 | 0.638 | 0.759 | 0.713 | 0.634 | 0.756 | 0.687 | 0.603 | 0.733 | 0.658 | 0.564 | 0.712 |
| MCLET-MHAM | **0.747** | **0.674** | **0.790** | **0.747** | **0.675** | **0.787** | **0.729** | **0.652** | **0.773** | **0.703** | **0.619** | **0.750** |

Table 6: Evaluation with different relation types dropping rates on FB15kET. H@N is an abbreviation for Hits@N, $N \in \{1, 3\}$. Best scores are highlighted in **bold**.

Relational neighbors of an entity provide supporting facts for its representation. To verify the robustness of MCLET in scenarios where relational neighbors are relatively sparse, we conduct an ablation experiment on FB15kET by randomly removing 25%, 50%, 75%, and 90% of the relational neighbors of entities, as proposed in (Hu et al., 2022a). We note that even after removing different proportions of relational neighbors, MCLET still achieves optimal performance. This can be mainly explained by two reasons. On the one hand, our views are based solely on the entity type neighbors, without involving the entity relational neighbors. Thus, changes in relational neighbors do not significantly affect the performance of MCLET. On the other hand, relational neighbors only serve as auxiliary knowledge for entity type inference, while the existing type neighbors of entities play a decisive role in predicting the missing types of entities.

**Effect of Dropping Rates of Relation Types.**
Compared with YAGO43kET, FB15kET has much more relations. To verify the robustness of MCLET when the number of relations is small, similar to (Hu et al., 2022a), we randomly remove 25%, 50%, 75%, and 90% of the relation types in FB15kET. From Table 6, we can observe that even with a smaller number of relation types, MCLET still achieves the best performance. This demonstrates the robustness of our method in the presence of significant variations in the number of relational

neighbors. This is mainly due to the introduction of cluster information, which establishes a coarse-grained bridge between entities and types, this information is not affected by drastic changes in the structure of the knowledge graph. Therefore, the incorporation of cluster information is necessary for entity type prediction tasks.

# 6 Conclusions

We propose MCLET, a multi-view contrastive learning (CL) framework for KGET. We design three different views with different granularities, and use a CL strategy to achieve cross-view cooperatively interaction. By introducing multi-head attention with a Mixture-of-Experts mechanism, we can combine different neighbor prediction scores. For future work, we plan to investigate inductive scenarios, dealing with unseen entities and types.

# Acknowledgments

This work has been supported by the National Natural Science Foundation of China (No.61936012, No.62076155), by the Key Research and Development Program of Shanxi Province (No.2021020201008), by the Science and Technology Cooperation and Exchange Special Program of Shanxi Province (No.202204041101016), by the Chang Jiang Scholars Program (J2019032) and by a Leverhulme Trust Research Project Grant (RPG-2021-140).

# 7 Limitations

In this paper, we introduce coarse-grained cluster content for the knowledge graph entity typing task. Although we achieve good results, there are still limitations in the following aspects: 1) For the standard benchmark datasets we use the readily available cluster-level annotation information. However, for those datasets without cluster information, we would need to use clustering algorithms to construct implicit cluster semantic structures. 2) There is a related task named fine-grained entity prediction (FET), the difference lies in predicting the types of entities that are mentioned in a given sentence, rather than entities present in a knowledge graph. The corresponding benchmarks also have annotated coarse-grained cluster information. Therefore, it would be worthwhile exploring the transferability of MCLET to the FET task.

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

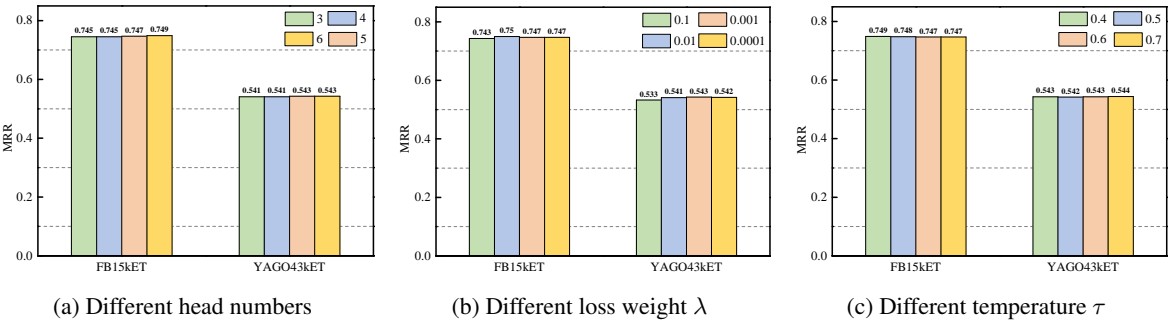

Figure 3: The ablation studies results under different experimental conditions.

# Appendix

## A Details about Experiments

| Parameter | {FB15kET, YAGO43kET} |
|---|---|
| # Embedding dimensions | {100, 100} |
| # Learning rate | {0.001, 0.001} |
| # Temperature | {0.6, 0.6} |
| # LightGCN layers | {4, 1} |
| # Number of heads | {5, 5} |
| # Number of experts | {32, 32} |
| # $\beta$ value | {4, 2} |
| # $\lambda$ value | {0.001, 0.001} |
| # $\gamma$ value | {1e-5, 1e-5} |

Table 7: The best hyperparameters of MCLET model in different datasets.

**Hyperparameter Settings.** All experiments were carried out on a 32G Tesla V100 GPU, we use Adam (Kingma and Ba, 2015) as the optimizer, and determine the final value of each hyperparameter based on the MRR value on the validation set by using grid search. We fine-tune the hyperparameters including the number of embedding dimensions from $d \in \{50, 100, 150, 200\}$, and the learning rate from $lr \in \{0.001, 0.005, 0.01\}$, the temperature parameter in contrastive loss $\tau \in \{0.4, 0.5, 0.6, 0.7\}$, the number of Light-GCN layers $L \in \{1, 2, 3, 4\}$, the number of heads $H \in \{3, 4, 5, 6\}$, the numbers of experts $M \in \{16, 32, 64\}$, the weight of negative samples in FNA loss $\beta \in \{1, 2, 3, 4\}$, the weight of contrastive loss $\lambda \in \{0.1, 0.01, 0.001, 0.0001\}$, the weight of $L2$ regularization $\gamma \in \{$1e-5, 2e-5, 3e-5$\}$. Table 7 summarizes the best configurations in two datasets.

## B Additional Results

**Effect of Different Head Numbers.** In Figure 3 (a) we report the results of testing the performance of different numbers of attention heads in MHAM.

We observe that the selection of different numbers of attention heads has a slight impact on the performance. It should be noted that, as shown in Equation 7, using more heads will introduce more learning parameters, so when the performance of the model is equivalent, it is better to use fewer heads.

**Effect of Different Loss Weight $\lambda$.** The parameter $\lambda$ in Equation 8 determines the importance of the contrastive loss during the multi-loss training process. We set the weight values of the contrastive loss as $\in \{0.1, 0.01, 0.001, 0.0001\}$ to investigate its impact on the final entity typing prediction results. From Figure 3 (b), one can see that different weight values have a minimal effect on the overall performance. This demonstrates the robustness of MCLET in terms of setting the weight for the contrastive loss, ensuring that the results do not undergo drastic changes due to improper weight values.

**Effect of Different Temperature $\tau$.** The temperature coefficient can be used to adjust the similarity measurement between samples. A higher temperature coefficient will flatten the distribution of similarities, causing the model to pay more attention to subtle differences between samples. On the other hand, a lower temperature coefficient will concentrate the distribution of similarities, emphasizing the overall differences between samples. From Figure 3 (c), it can be observed that setting different temperature coefficients has limited impact on MCLET. Although the choice of temperature coefficient usually requires adjustment based on specific tasks and datasets, in our current model context, the selection of this coefficient does not have a decisive effect on the final performance of the model.

## C  Complexity Analysis

| Model | FB15kET | YAGO43kET |
|-------|---------|-----------|
| CompGCN (Vashishth et al., 2020) | 98.395M | 604.690M |
| RGCN (Pan et al., 2021) | 45.875M | 578.330M |
| CET (Pan et al., 2021) | 504.627M | 6.362G |
| TET (Hu et al., 2022a) | 2.016G | 8.406G |
| MiNer (Jin et al., 2022) | 1.127G | 14.204G |
| MCLET | 1.129G | 13.620G |

Table 8: The amount of calculations required by different models on different datasets.

**Computational Complexity.**  Considering that the prediction accuracy of embedding-based methods for KGET is relatively lower, we compared the computational complexity of MCLET with GNN-based models like CompGCN, RGCN, CET, TET, and MiNer, the corresponding results are shown in Table 8. The computational cost refers to the number of floating-point operations (FLOPs) required during training or inference.  We incur greater computational overhead compared to CompGCN, RGCN, and CET, but the performance of these three baselines is significantly lower than that of MCLET. In comparison to the SoTA models TET and MiNer, we require less computational resources than TET on the FB15kET dataset, and than MiNer on the YAGO43kET dataset. This indicates that we can achieve optimal performance without introducing a substantial computational overhead. Additionally, note that for the FB15kET and YAGO43kET datasets, the MCLET model respectively converges in only 39 minutes and 7 hours 10 minutes, for a Tesla V100 32G graphics card, which is a reasonable time cost.