# OpenReview forum: "Multi-view Contrastive Learning for Entity Typing over Knowledge Graphs"
_EMNLP/2023/Conference — EMNLP 2023 Main_

### Official Review · Reviewer_sir8 · 2023-08-05

**Soundness:** 4

**Excitement:**

4: Strong: This paper deepens the understanding of some phenomenon or lowers the barriers to an existing research direction.

**Paper Topic And Main Contributions:**

This paper proposes a knowledge graph entity typing approach by leveraging entity-type, entity-cluster and cluster-type structured information. A cross-view contrastive learning module is designed to capture the interaction between different views. A multi-head attention with Mixture-of-Experts mechanism is introduced to distinguish the contribution from different entity neighbors.

**Questions For The Authors:**

1. Can you please compare the computational complexity of different approaches for entity typing?
2. Need more introduction and explanation of LightGCN
3. At the bottom of page 4, please give a better explanation of reason behind adding up graph convolutional layers.
4. Please explain more clearly how to find positive samples and negative samples, perhaps with a figure illustration.


**Reasons To Accept:**

Clearly discuss type knowledge clustering, entity type graph, cluster type graph, entity cluster graph and offer sensible examples for each of the definition. Introducing LightGCN instead of RGCN as multi-view encoder and clearly explain the choice of encoder. This paper also introduce contrastive learning loss that considers both intra-view and inter-view negative samples, and making the contrastive learning loss working together with false-negative aware loss function. Predicting type by propagating neighbor type information is also very logical. The robustness of Multi-head attention with a mixture of experts also backed by results comparison of dropping relation neighbors. This paper conducts ample experiments and shows results of different variants and plenty of ablation studies. Presentation of results in tables are also very clear. Overall this paper is well written and shows clear effort and thoughts into presentation and organization.

**Reasons To Reject:**

My only concern is the model might be overly complex and how applicable is it to type prediction for large knowledge graphs, and whether this increase in complexity brings enough benefit.

**Reproducibility:**

4: Could mostly reproduce the results, but there may be some variation because of sample variance or minor variations in their interpretation of the protocol or method.

**Reviewer Confidence:**

5: Positive that my evaluation is correct. I read the paper very carefully and I am very familiar with related work.

**Typos Grammar Style And Presentation Improvements:**

The subscript notation is not very clearly explained in the paper which affects understanding. All subscript notations need better explanation. The part for how to select positive and negative samples need to be reworded.

---

> ### Author Rebuttal · Authors · 2023-08-27
>
> >**Q1:** *My only concern is the model might be overly complex and how applicable is it to type prediction for large knowledge graphs, and whether this increase in complexity brings enough benefit.*
>
> **Answer:**
> 1. Regarding the analysis of computational complexity, we compared our model with the five baselines, see details in Q2 below.
>     - On the one hand, compared to the SoTA models TET and MiNer, MCLET did not introduce any additional computational overhead. It even required less overhead on the YAGO43kET dataset than the MiNer model, and less overhead on the FB15kET dataset than TET model.
>
>     - On the other hand, with respect to the MRR metric, on both the FB15kET and YAGO43kET datasets, MCLET outperformed MiNer, achieving improvements of 2.2% in each.  From the computational complexity perspective, these results show the feasibility of migrating MCLET to larger scale KGs. Of course, during the migration process, obtaining information about clusters of types is essential.
> 2. In addition, we investigate the convergence time of MCLET on the FB15kET and YAGO43kET datasets. For a Tesla V100 32G graphics card, MCLET respectively takes 39 minutes and 7 hours 10 minutes for model convergence on the FB15kET and YAGO43kET datasets. So, the cost of running the model is acceptable.
>
> >**Q2:** *Can you please compare the computational complexity of different approaches for entity typing?*
>
> **Answer:**
> - The computational cost refers to the number of floating-point operations (FLOPs) required during training or inference. We compared the computational complexity of CompGCN, RGCN, CET, MiNer, TET and MCLET on the FB15kET and YAGO43kET datasets, as shown in the table below.
>     - For the FB15kET dataset, compared to the SoTA  models TET and  MiNer models, MCLET has a lower number of FLOPs than TET and on par with the MiNer model. Although CompGCN, RGCN and CET models have smaller FLOPs values on the FB15kET dataset, the accuracy of these models is not of the same order as that of TET, MiNer and MCLET.
>     - For the YAGO43kET dataset, MCLET has lower number of FLOPs than MiNer. However, compared with TET, MCLET requires more computational overhead, but considering that the MRR metric is 3.3% higher than TET, it is a tolerable increase.
>     - To sum up, MCLET can achieve a relatively large performance improvement with a computational cost that is not much different from SoTA, so we believe the design of the model is reasonable.
>
>
> Model | FB15kET | YAGO43kET
> | :--: | :--: | :--:
> CompGCN | 98.395M | 604.690M
> RGCN | 45.875M | 578.330M
> CET | 504.627M | 6.362G
> TET | 2.016G | 8.406G
> MiNer | 1.127G | 14.204G
> MCLET | 1.129G | 13.620G
>
> >**Q3:** *Need more introduction and explanation of LightGCN.*
>
> **Answer:**
> - We start by noticing that we considered LightGCN solely as an encoder for different views. Since it is not the focus of our work and due to space restrictions, we did not provide a very detailed introduction of LightGCN. We plan to include a detailed introduction to LightGCN in the final version, as there will be an additional page available.
>
> >**Q4:** *At the bottom of page 4, please give a better explanation of reason behind adding up graph convolutional layers.*
>
> **Answer:**
> 1. We first note that the practice of summing and averaging the stacked layers encoded by LightGCN is a popular paradigm. We provide a few references here as examples: [1-4]. We will include them in the final version.
>
>     - [1] LightGCN: Simplifying and Powering Graph Convolution Network for Recommendation. SIGIR, 2020.
>     - [2] Improving Graph Collaborative Filtering with Neighborhood-enriched Contrastive Learning, WWW, 2022.
>     - [3] Knowledge Graph Self-Supervised Rationalization for Recommendation, KDD, 2023.
>     - [4] Multi-Behavior Recommendation with Cascading Graph Convolution Networks, WWW, 2023.
>
> 2. Furthermore, based on the FB15kET dataset, we conducted ablation experiments involving the encoding results of different layers of LightGCN for subsequent operations, see table below. It's worth mentioning that the main reason we did not perform experiments on the YAGO43kET dataset is that, for this dataset, the number of layers chosen for LightGCN was 1 (cf. Tables 6 and 3in the appendix of the original manuscript). Therefore, conducting ablation experiments on different layers was not necessary for YAGO43kET.
>
> Setting | MRR | Hits@1 | Hits@3 | Hits@10
> | :--: | :--: | :--: | :--: | :--: |
> mean-all-layers (1-4 layers) | 0.750 | 0.677 | 0.793 | 0.891
> mean-middle-layers (2-3 layers) | 0.744 | 0.676 | 0.778 | 0.881
> max-all-layers (1-4 layers) | 0.748 | 0.677 | 0.788 | 0.887
> max-middle-layers (2-3 layers) | 0.745 | 0.675 | 0.783 | 0.881
> last-layer | 0.746 | 0.675 | 0.785 | 0.888
>
> 3. For the FB15kET dataset, we set the number of layers for LightGCN to 4 and considered two fusion methods for different layers: taking the mean and taking the maximum value. We investigated the experimental results under the conditions of all-layers, middle-layers, and  last-layer. As there is only one layer in the last layer condition, no mean or max operations are needed. It was evident that the best experimental results were achieved under the all-layers condition, and the mean operation approach exhibited a slight advantage over the max operation. However, generally speaking, the performance difference between mean and max is not very large, so the chosen strategy has no decisive impact on the final model results. The results reported in our original manuscript were based on the mean operation applied to all-layers.
>
> >**Q5:** *Please explain more clearly how to find positive samples and negative samples, perhaps with a figure illustration.*
>
> **Answer:**
> - Unfortunately, we can't show how the positive and negative samples are obtained through a figure, which would be the most adequate.  We next provide an explanation. We adopted a method for obtaining positive and negative samples consistent with the CET, TET and MiNer baselines.
>     - For positive samples, the known (entity, type) pairs in the training set can be used as positive samples. For example, if (Joe Biden, American_politician), (Barack Obama, male_lawyer), and (Joe Biden, male_politician) exist in the training set, then these three tuple pairs can be used as positive samples.
>     - To gather the negative samples, a simple choice is to treat all (entity, type) pairs that do not occur in the training set as negative samples. However, some (entity, type) pairs are valid but happen to be missing in current knowledge graphs due to their incomplete nature. So, considering them as negative samples would introduce a serious false-negative phenomenon during training. For example, suppose the training set contains the tuples (Joe Biden, American_politician), (Barack Obama, male_lawyer), and (Joe Biden, male_politician). When considering negative samples for (Joe Biden, American_politician), we might obtain (Joe Biden, male_lawyer) as a negative example, because it doesn't exist in the knowledge graph. However, in reality, (Joe Biden, male_lawyer) should be a positive example but is incorrectly considered negative due to the incompleteness of the knowledge graph, causing what is known as the *"false negative”* phenomenon. We mentioned in  the *"Prediction and Optimization"* part of the submision that FNA can be used to solve the false negative problem. We are consistent with CET, TET and MiNer on this point.
>
> >**Q6:** *The subscript notation is not very clearly explained in the paper which affects understanding. All subscript notations need better explanation. The part for how to select positive and negative samples need to be reworded.*
>
> **Answer:**
> - For the ambiguity of the subscript, we will make corrections in the final version. For the problem that the positive and negative sample selection process is not described, we will add some examples in the final version to explain in detail.

---

### Official Review · Reviewer_sXvZ · 2023-08-05

**Soundness:** 3

**Excitement:**

3: Ambivalent: It has merits (e.g., it reports state-of-the-art results, the idea is nice), but there are key weaknesses (e.g., it describes incremental work), and it can significantly benefit from another round of revision. However, I won't object to accepting it if my co-reviewers champion it.

**Paper Topic And Main Contributions:**

The paper targets entity type prediction in KGs. It proposes multi-view modeling of entity neighbouring information and a mixture of multiple aspects to make the final prediction and achieve better results.

**Questions For The Authors:**

1. Can your model handle unseen entities?

2. The proposed method consists of multiple modules and I have seen similar modules from different applications. Which newly proposed modules do you think are the most important to the overall framework?

**Reasons To Accept:**

1. The idea of incorporating multi-view information is intuitive. It divides KG into several aspects and handles them separately. Finally, all information is fused to make the final decision for entity types.

2. Good performance is shown compared to previous models.

**Reasons To Reject:**

1. I am not sure familiar with all previous models on entity tying. It would be a bit shock to me if the previous methods have never considered multi-view information aggregation of KGs. If a similar concept exists, how the methods and motivations are different from the current manuscript?

2. The proposed method includes a couple of modules and builds up a quite complex model. How efficient is the proposed method compared to the previous method? Especially compared with GNN-based and embedding-based methods. A complexity analysis would be very helpful.

**Reproducibility:**

4: Could mostly reproduce the results, but there may be some variation because of sample variance or minor variations in their interpretation of the protocol or method.

**Reviewer Confidence:**

3: Pretty sure, but there's a chance I missed something. Although I have a good feel for this area in general, I did not carefully check the paper's details, e.g., the math, experimental design, or novelty.

---

> ### Author Rebuttal · Authors · 2023-08-27
>
> >**Q1:** *I am not sure familiar with all previous models on entity tying. It would be a bit shock to me if the previous methods have never considered multi-view information aggregation of KGs. If a similar concept exists, how the methods and motivations are different from the current manuscript?*
>
> **Answer:**
> 1. To our knowledge, existing models have not adopted a multi-view approach for the KGET task. The primary reason is that the existing baseline models have not incorporated cluster knowledge. So, different perspectives of knowledge are not formed. By introducing cluster information, we established three levels of information: entity, coarse-grained cluster, and fine-grained type. We can construct a view by considering only two levels, and then the three different views can be formed by combining the three in pairs. However, current methods only consider information at the entity and fine-grained type level.
> 2. In a nutshell, the main innovation points of our submission are:
>     - Firstly, we introduced coarse-grained type cluster knowledge into the KGET task for the first time, creating three distinct views: entity-type, cluster-type, and entity-cluster, and different views can capture content at different levels of granularity.
>     - Then, we use a cross-view contrastive learning mechanism to capture multi-grained information to obtain better embedding representations.
>     - Finally, we devise a multi-head attention with a Mixture-of-Experts strategy to distinguish the contribution from different neighbors.
>
> >**Q2:** *The proposed method includes a couple of modules and builds up a quite complex model. How efficient is the proposed method compared to the previous method? Especially compared with GNN-based and embedding-based methods. A complexity analysis would be very helpful.*
>
> **Answer:**
> 1. Considering that the prediction accuracy of embedding-based methods for KGET is low, we compared the computational complexity of MCLET with the GNN-based models like CompGCN, RGCN, CET, TET, and MiNer, see the table below. The computational cost refers to the number of floating-point operations (FLOPs) required during training or inference. We incur greater computational overhead compared to CompGCN, RGCN, and CET, but the performance of these three baselines is significantly lower than that of MCLET. In comparison to the SoTA models TET and MiNer, we require less computational resources than TET on the FB15kET dataset, and than MiNer on the YAGO43kET dataset. This indicates that we can achieve optimal performance without introducing a substantial computational overhead.
> Model | FB15kET | YAGO43kET
> :--: | :--: | :--:
> CompGCN | 98.395M | 604.690M
> RGCN | 45.875M | 578.330M
> CET | 504.627M | 6.362G
> TET | 2.016G | 8.406G
> MiNer | 1.127G | 14.204G
> MCLET | 1.129G | 13.620G
> 2. Additionally, note that for the FB15kET and YAGO43kET datasets, the MCLET model respectively converges in only 39 minutes and 7 hours 10 minutes, for a Tesla V100 32G graphics card, which is a reasonable time cost.
>
> >**Q3:** *Can your model handle unseen entities?*
>
> **Answer:**
> - No, MCLET cannot handle unseen entities. Not only MCLET but also all baselines, like CET, TET, and MiNer, are unable to handle unseen entities (in other words, the inductive scenario). Indeed, there are not KGET task datasets that cater to this inductive condition. This is a promising direction for exploration, and we intend to further develop this aspect in our future work.
>
> >**Q4:** *The proposed method consists of multiple modules and I have seen similar modules from different applications. Which newly proposed modules do you think are the most important to the overall framework?*
>
> **Answer:**
> - In terms of the impact on model performance, the *"Multi-view Generation and Encoder"* and *"Cross-view Contrastive Learning"* are the most important. This follows from the ablation experiments shown in Table 2. The experiments demonstrated that removing all views led to a significant performance drop, both on the FB15kET and YAGO43kET datasets. This aligns with the initial motivation of introducing type cluster knowledge to the KGET task. Indeed, these results confirm the importance of constructing multiple views and encoding them after incorporating cluster knowledge. Furthermore, the embeddings obtained from different views capture  specific information of each view, without considering interactions between embeddings across views. We addressed this by introducing a cross-view contrastive learning mechanism, which facilitated mutual awareness between different view contents. This enhancement led to more discriminative embedding representations.

---

### Official Review · Reviewer_Hx7N · 2023-08-05

**Soundness:** 4

**Excitement:**

3: Ambivalent: It has merits (e.g., it reports state-of-the-art results, the idea is nice), but there are key weaknesses (e.g., it describes incremental work), and it can significantly benefit from another round of revision. However, I won't object to accepting it if my co-reviewers champion it.

**Paper Topic And Main Contributions:**

This paper proposed a new approach, named MCLET, to predict missing entity types in knowledge graphs (KGs). Relationships between entities, coarse-grained, and fine-grained types are considered. Three modules are included in MCLET to process multi-view KGs, cross-view KGs, and entity type prediction, respectively.

**Questions For The Authors:**

1. How are the clusters of types generated or obtained? To my knowledge, YAGO doesn't contain such information.

**Reasons To Accept:**

1. It's novel to consider coarse-to-fine entity types through type clusters.
2. The content is solid, and the experiments are thorough and complete.

**Reasons To Reject:**

1. It's unclear how the types are clustered (see questions below).
2. It doesn't make sense to me how the entity-entity relationships are not considered and missing in the model.

===== After Rebuttal =====

Thanks to the author for the detailed response. It addressed most of my questions and concerns. However, I decided not to change the current scores.

**Reproducibility:**

4: Could mostly reproduce the results, but there may be some variation because of sample variance or minor variations in their interpretation of the protocol or method.

**Reviewer Confidence:**

3: Pretty sure, but there's a chance I missed something. Although I have a good feel for this area in general, I did not carefully check the paper's details, e.g., the math, experimental design, or novelty.

---

> ### Author Rebuttal · Authors · 2023-08-27
>
> >  **Q1**: *It's unclear how the types are clustered or how are the clusters of types generated or obtained? To my knowledge, YAGO doesn't contain such information.*
>
> **Answer:**
> 1. For the FB15kET dataset, we can directly obtain cluster information using a rule-like approach based on their type annotations. For instance, the type *"/location/uk_overseas_territory"* belongs to the cluster *"location"* and the type *"/education/educational_degree"* belongs to the cluster *"education"*.
> 2. For the YAGO43kET dataset, as mentioned in the submission, it provides an alignment between types and WordNet concepts. So, we can directly obtain the words in WordNet describing the cluster to which a type belongs to. For example, for the type *"wikicategory_People_from_Dungannon"*, its cluster is *"wordnet_person_100007846"*, and for the type *"wikicategory_Male_actors_from_Arizona"*, its cluster is *"wordnet_actor_109765278"*.
> 3. Furthermore, one can find the following description on the official website of the YAGO dataset: *"YAGO classifies each entity into a taxonomy of classes. Every entity is an instance of one or multiple classes. Every class (except the root class) is a subclass of one or multiple classes. This yields a hierarchy of classes — the taxonomy. The taxonomy consists of 4 layers."* The introduction to the third layer is as follows: *"The middle layer of the taxonomy consists of classes that have been derived from Wikipedia categories. For example, one class is <wikicategory_American_rock_singers>, derived from the Wikipedia category American rock singers. Each of these classes is connected to one class of the WordNet layer by a rdfs:subclassOf relationship. In the example, <wikicategory_American_rock_singers> rdfs:subclassOf <wordnet_singer_110599806>."* So, at this layer every type from Wikipedia can be aligned with a category from WordNet. This is the data source for us to obtain the type cluster information in the YAGO dataset.
>
> >  **Q2**: *It doesn't make sense to me how the entity-entity relationships are not considered and missing in the model.*
>
> **Answer:**
> 1. Note that we did not directly utilize the entity-entity graph, but we indirectly incorporated information from the entity-entity graph into the *"Neighbor Prediction Mechanism"* of the *"Entity Typing Prediction"* module. In this mechanism, we utilize the relational neighbors and type neighbors of entities to predict missing entity types. The relational neighbors of entities are derived from the entity-entity graph.
> 2. There are two main reasons to not directly employ the entity-entity graph.
>     - On the one hand, as discussed in the *"Task Definition"*, a knowledge graph comprises two subgraphs: $\mathcal G_{\textit{triples}}$ and $\mathcal G_{\textit{types}}$. The former corresponds to the entity-entity graph you referred to, while the latter represents the entity-type graph. Given the goal of the KGET task, our focus is on finding ways to better utilize the entity-type graph. We introduced type cluster information into the entity-type graph, constructing three views: entity-type, cluster-type, and entity-cluster, and employed cross-view contrastive learning to obtain better embeddings.
>     - On the other hand, the entity-entity graph is a multi-relational graph, encoding it would require a computation-intensive encoder such as RGCN. In contrast, the three views we acquired (entity-type, cluster-type, and entity-cluster) are single-relational graphs. For them, a lightweight encoder like LightGCN suffices. This choice ensures performance enhancement without introducing excessive computational burden.
> 3. In addition, we have conducted a computational cost analysis of MCLET compared to CompGCN, RGCN, CET, TET, and MiNer models on the FB15kET and YAGO43kET datasets (as shown in the table below). The computational cost refers to the number of floating-point operations (FLOPs) required during training or inference. We can observe that, when compared to the SoTA models TET and MiNer, MCLET exhibits lower computational costs on the FB15kET dataset than TET and on the YAGO43kET dataset than MiNer.
>
>
> Model | FB15kET | YAGO43kET
> | :--: | :--: | :--: |
> CompGCN | 98.395M | 604.690M
> RGCN | 45.875M | 578.330M
> CET | 504.627M | 6.362G
> TET | 2.016G | 8.406G
> MiNer | 1.127G | 14.204G
> MCLET | 1.129G | 13.620G

---

### Meta-Review · Area_Chair_Jcaj · 2023-09-18

**Recommendation:** 4

**Metareview:**

This paper approaches the problem of entity typing and proposes a method that leverages entity-type, entity-cluster, and cluster-type information. Reviewers praised the papers clarity, thoroughness of experiments as well as the design and performance of the method. The main concerns for the work were whether the additional complexity of the method was justified by the amount of performance increase tradeoffs and if the methods were sufficiently different from prior approaches.

---

### Decision · Program_Chairs · 2023-10-07

**Decision:**

Accept-Main

**Comment:**

This paper approaches the problem of entity typing and proposes a method that leverages entity-type, entity-cluster, and cluster-type information. Reviewers praised the papers clarity, thoroughness of experiments as well as the design and performance of the method. The main concerns for the work were whether the additional complexity of the method was justified by the amount of performance increase tradeoffs and if the methods were sufficiently different from prior approaches.